# Socio-demographic determinants of Monkeypox virus preventive behavior: A cross-sectional study in Pakistan

Hashaam Jamil[1], Muhammad Idrees[2], Kashmala Idrees[1], Waleed Tariq[1], Qudsia Sayyeda[3], Muhammad Sohaib Asghar[4], Muhammad Junaid Tahir[5], Samra Akram[1], Kaleem Ullah[6], Ali Ahmed[7], Aroop Mohanty[8], Bijaya Kumar Padhi[9], Ranjit Sah[10,11,12]*

1 Lahore General Hospital, Lahore, Pakistan, 2 Multan Institute of Kidney Diseases, Multan, Pakistan, 3 Red Crescent of Tampa Bay, Tampa, FL, United States of America, 4 Mayo Clinic, Rochester, MN, United States of America, 5 Pakistan Kidney and Liver Institute and Research Center, Lahore, Pakistan, 6 Pir Abdul Qadir Shah Jeelani Institute of Medical Sciences, Gambat, Pakistan, 7 School of Pharmacy, Monash University, Subang Jaya, Selangor, Malaysia, 8 Department of Microbiology, All India Institute of Medical Sciences, Gorakhpur, India, 9 Department of Community Medicine and School of Public Health, Postgraduate Institute of Medical Education and Research, Chandigarh, India, 10 Tribhuvan University Teaching Hospital, Institute of Medicine, Kathmandu, Nepal, 11 Department of Microbiology, Dr. D. Y. Patil Medical College, Hospital and Research Centre, Dr. D. Y. Patil Vidyapeeth, Pune, Maharashtra, India, 12 Harvard Medical School, Boston, MA, United States of America

* ranjitsah@iom.edu.np

**Data Availability Statement:** All relevant data are within the manuscript and its Supporting Information files.

## Abstract

### Background

Monkeypox (mpox), re-emerging zoonotic infectious disease, is striking the world with serious public health concerns, especially in non-endemic countries. The public's knowledge and attitude towards the monkeypox virus (MPXV) influence their adherence to preventive strategies. Therefore, we aimed to assess the public's knowledge, attitudes, and perceptions (KAP) of MPXV in Pakistan.

### Methods

We collected data for this cross-sectional study from 1040 participants via online self-reported questionnaire from July 5th, 2022, to August 1st, 2022. The questionnaire consisted of a total of 29 items in four sections, assessing socio-demographics, knowledge, attitudes, and practices regarding MPXV. The data were analyzed using IBM SPSS V.25, and factors associated with MPXV knowledge, attitude, and practices were identified by using logistic regression analyses.

### Results

A total of 1040 participants were included. 61.4% were male, and 57.2% had graduation level education. Only 34.4% had good knowledge about MPXV, and 30% knew the effectiveness of the smallpox vaccine against MPXV. 41.7% had a positive attitude, 48.6% agreed that it is a fatal disease, and 44.6% were in favour of banning travel from endemic to

**Funding:** The author(s) received no specific funding for this work.

**Competing interests:** The authors have declared that no competing interests exist.

non-endemic regions. 57.7% had good practices, and 69.9% would use protective measures if MPXV became an epidemic. Binary logistic regression analysis revealed that gender and education significantly impacted knowledge (p<0.05). While monthly income status had a significant impact on attitudes (p<0.05). The practices were positively correlated with gender and education (p<0.05).

## Conclusion

The majority of study participants had inadequate levels of knowledge, and attitudes regarding MPXV. To prevent the emergence and spread of MPXV in Pakistan, a comprehensive strategic framework for public health education must be established and implemented.

## Introduction

The monkeypox virus (MPXV), a member of the *POXVIRIDAE* family of the genus *Orthopoxvirus*, is the pathogen responsible for monkeypox (mpox) fever, a zoonotic infectious disease [1]. The first known outbreak of the virus in monkeys was discovered in 1958 at an animal research centre in Copenhagen, Denmark, giving rise to the term "monkeypox" [2]. MPXV was not recognized as a human virus until 1970 when the virus was isolated from a child in the Democratic Republic of the Congo (DRC) who had symptoms of suspected smallpox infection [3]. The potential modes of MPXV transmission are either animal-human or human-human. Animal-human transmission is associated with direct contact with the blood, body fluids, and cutaneous or mucosal wounds of infected animal hosts [4]. Close contact with respiratory secretions, infected patient objects or surroundings, an infected person's skin lesions, and sexual contact with an infected person who has genital and groin lesions can cause human-to-human transmission [5]. However, in the recent multi-country MPXV outbreak, the majority of MPXV cases have primarily been reported among men who have had sex with men (MSM) [6–8].

Human mpox has clinical features that are strikingly similar to those of smallpox, but less severe [9, 10]. After an incubation period of 10 to 14 days, the majority of patients experience prodromic symptoms, including fever, malaise, and enlarged lymph nodes. About 90% of patients infected with MPXV develop lymphadenopathy [11, 12]. After an initial prodromal period, a centrifugally distributed maculopapular rash developed. Lesions develop from macules to papules, vesicles, pustules, and eventually crust over in two to four weeks [13]. Treatment of MPXV is mainly symptomatic. The smallpox vaccination is 85% effective in protecting against mpox [14]. However, Modified Vaccinia Ankara-Bavarian Nordic (MVA-BN), an authorized vaccine, and the antiviral medications tecovirimat and brincidofovir are not easily available [15, 16].

In 2003, the United States reported the first MPXV outbreak in humans outside of Africa, with 70 cases, which was associated with the importation of exotic pets from Ghana [17]. Since then, the incidence in endemic areas has substantially increased, and sporadic, travel-related incidents are on exponential rise in non-endemic nations [13, 18]. In September 2017, Nigeria reported a major outbreak of human MPXV, with about 228 suspected and 60 confirmed cases caused by the West African clade [19]. In the ongoing outbreak, the UK Health Security Agency reported the first mpox case on May 07, 2022. Since that time, the World Health Organization (WHO) has consistently recorded reports of mpox cases [20, 21]. Mpox was declared a public health emergency of international concern by the WHO on July 23, 2022, due to an

upsurge in cases [22]. From January 1[st] through November 17[th], 2022, 80221 confirmed cases of MPXV via PCR and 52 mortalities have been reported to the WHO from 110 countries [23]. Although MPXV has been reported in other southeast Asian countries (India, Thailand, and Indonesia) [24, 25], so far, Pakistan has not reported any cases of MPXV. Although, there were reports on two rarely occurring cases of the zoonotic mpox infection [26], but the reporting of mpox cases in Pakistan was false news, which was refuted by the National Institute of Health on May 24, 2022 [27].

According to the WHO, a lack of awareness about the virus is one of the obstacles in preventing the re-emergence of MPXV [28]. Adherence to prevention and control initiatives is important in restricting the dissemination of the virus, which is largely influenced by the community' cumulative knowledge, attitudes, and practices (KAP) [29, 30]. A comprehensive strategic framework must be established and implemented to prevent the emergence and spread of MPXV in Pakistan. Our findings emphasize the need to investigate the KAP towards MPXV among the general population of Pakistan. The study's conclusions could assist public health officials in strengthening policies, increasing public awareness of the MPXV outbreak, and organizing future public health initiatives.

## Methods

### Study design, setting and data collection

An analytical cross-sectional study was applied to assemble data from Pakistan's general population using a non-probability convenience sampling technique. Using shared Google forms links across WhatsApp, Instagram, Facebook, and Twitter, Pakistan's most widely used and accessible social media platforms, the research team developed a questionnaire with informed consent. Between July 5[th], 2022, to August 1[st], 2022, data were collected across all of Pakistan's provinces (i.e., Sindh, Punjab, Baluchistan, and Khyber Pakhtunkhwa). Individuals below the age of 18, non-Pakistanis, and those who were unable to understand the questionnaire language were excluded from the study.

### Sample size

Pakistan's population, according to the most recent census, is 207.7 million [31]. Using a Raosoft sample size calculator online, minimum sample size of 752 was calculated using a 90% confidence interval, 50% of the population, a 3% margin of error, and a total population of 207.774 million. We ultimately obtained 1040 responses to include in the survey, which will help us get more accurate and reliable results.

### Questionnaire development

The self-reported questionnaire was developed using an extensive literature review [16, 32, 33] and information was sorted out from the WHO and Centers for Disease Control and Prevention's official websites (CDC) [34, 35]. This survey form was reviewed by a team comprising senior community medicine specialists, public health experts, and infectious disease researchers to improve some details for the pseudo-validity, relevance, completeness, and clarity of individual sections. After a detailed discussion and audit, the authors finalized the survey. Then, 80 people participated in pilot research to examine the reliability of the system. Cronbach's alpha was reported to be 0.76, and the results of the pilot study were not included in the final analysis.

The beginning of questionnaire was introduced with a brief overview of the nature and purpose of the study, informed consent, and then segregated into four divisions to assess the

socio-demographics of the participants, knowledge, attitudes, and practices. Six questions regarding age, gender, marital status, residence, education, and monthly income were included in the demographic section.

The second section evaluated the knowledge of participants of mpox by asking 10 questions. Each score was graded as "1" (correct) and "0" (wrong) with the scores ranging from 0 to 10. The study participants who scored 7 or more were classified as having good knowledge.

The third section assessed attitudes. Seven questions were tested by a 5–point Likert scale as follows: 1 ("strongly disagree"), 2 ("disagree"), 3 ("neutral"), 4 ("agree"), and 5 ("strongly agree"). The overall score for the attitude section ranged from 7 to 35, with an overall higher score depicting a more positive attitude. A score higher than or equal to 26 was described as a positive attitude.

The last section assessed the practices. Six questions were tested with a 5–point Likert scale as follows: 1 ("strongly disagree"), 2 ("disagree"), 3 ("neutral"), 4 ("agree"), and 5 ("strongly agree"). The practice section score ranged from 6 to 30, with an overall higher score depicting a more positive practice. A score more than or equal to 22 was considered a positive practice.

## Ethical considerations

The anonymity and confidentiality of participants were ensured during the survey. The research did not include names or other personal information. Before beginning the questionnaire, informed consent was taken from the sampled study participants. During the process of survey, participants can deny and withdraw at any time before submission. The ethical review committee of the Pir Abdul Qadir Shah Jeelani Institute of Medical Sciences in Gambat, Pakistan, approved the study protocol (Ref. No: IRB/22/13).

## Statistical analysis

The statistical package for social sciences (SPSS) version 25 was used for data analysis (IBM). While categorical variables were expressed as frequencies and percentages, numerical variables were quantified as means and standard deviations. Based on the variables and the data type, inferential statistics were used. Univariate logistic regression analysis carried out a comparison of knowledge, attitude, and practices with socio-demographic variables. The statistical results were demonstrated as odds ratio (OR) with a 95% confidence interval (CI) and p-value. In multivariable analysis, all with p-value <0.25 were included, and an adjusted odds ratio (aOR) was reported with independent association. A p-value of <0.05 was considered statistically significant.

## Results

### Socio-demographic characteristics

A total of 1040 participants were involved in the final analysis. Of the total, 61.4% (n = 639) were male, 61.8% (n = 643) fell in the age group 21–30 years, 79.5% (n = 827) were single. 79.5% (n = 789) of the respondents were residents of urban areas, and 57.2% (n = 595) achieved graduation level education (Table 1).

### General knowledge, attitudes, and practices about Mpox

This study determined that 34.4% (n = 358) of the participants had good knowledge, while 65.6% (n = 682) had poor knowledge about MPXV. 47.7% (n = 496) respondents correctly answered that MPXV was first isolated in Congo, and 87% (n = 905) correctly identified the causative agent of mpox is virus. 65% (n = 683) knew about preventions and 64% (n = 668)

**Table 1. Study sample socio-demographic characteristics (N = 1040).**

| Variables | Categories | Frequency | Percentage |
|---|---|---|---|
| Gender | | | |
| | Male | 639 | 61.4 |
| | Female | 401 | 38.6 |
| Age (years) | | | |
| | <20 | 234 | 22.5 |
| | 21–30 | 643 | 61.8 |
| | 31–40 | 124 | 11.9 |
| | >40 | 39 | 3.8 |
| Marital Status | | | |
| | Married | 213 | 20.5 |
| | Unmarried | 827 | 79.5 |
| Residence | | | |
| | Urban | 789 | 75.9 |
| | Rural | 251 | 24.1 |
| Education | | | |
| | Primary up to Matriculation | 38 | 3.7 |
| | Intermediate | 197 | 18.9 |
| | Graduation | 595 | 57.2 |
| | Post-graduation | 210 | 20.2 |
| Monthly incomein PKR (USD) | | | |
| | <50000 (225) | 644 | 61.9 |
| | 50000–100,000 (225–450) | 252 | 24.2 |
| | >100,000 (450) | 144 | 13.8 |

PKR: Pakistani Rupee.

were familiar with the symptoms. Only 30% (n = 312) of participants were familiar with the fact that the Smallpox vaccine is also effective against MPXV (Table 2).

Regarding attitudes, 41.7% (n = 434) of the participants had positive attitudes. 48.6% (n = 505) agreed that MPXV is a fatal disease, and 38.3% (n = 398) agreed that MPXV can cause a pandemic like the Coronavirus disease 2019 (COVID-19). 44.6% (n = 464) believed that traveling from endemic areas (such as Africa, Nigeria, etc.) to non-endemic areas should be banned (Table 3).

Assessment of practices showed that the majority of participants, 57.7% (n = 600), had good practices. About half (48.0%) of respondents agreed that infected patients should be isolated from others. 38.9% (n = 405) strongly agreed that there is a need for more awareness of MPXV in the general public. When asked whether the government of Pakistan is taking adequate steps to tackle MPXV, 27.0% (n = 281) were neutral to making the decision (Table 4).

## Factors of good knowledge, attitudes, and practices

The regression analysis revealed that men had a higher odds ratio than women [Adjusted odds ratio (aOR) = 1.324; 95% Confidence interval (CI) = 1.001–1.752, p = 0.049]. Participants with an education level from primary up to matriculation (Class-10) were 0.22 times less likely to have good knowledge than those with intermediate education (Class-12) (aOR = 0.220; 95% CI = 0.074–0.653, p = 0.006) (Table 5).

With respect to the different variables affecting attitudes regarding MPXV, and based on statistical values obtained by multivariate logistic regression, participants with an income of

**Table 2. Knowledge of general population regarding Mpox virus (N = 1040).**

| Variables | Categories | Frequency | Percentage |
|---|---|---|---|
| Mpox disease was first isolated in? | | | |
| | Incorrect | 544 | 52.3 |
| | Correct | 496 | 47.7 |
| The causative factor of mpox disease is? | | | |
| | Incorrect | 135 | 13.0 |
| | Correct | 905 | 87.0 |
| Transmission of mpox disease caused by? | | | |
| | Incorrect | 491 | 47.2 |
| | Correct | 549 | 52.8 |
| Human-to-human transmission caused by? | | | |
| | Incorrect | 554 | 53.3 |
| | Correct | 486 | 46.7 |
| The incubation period (interval from infection to onset of symptoms)is? | | | |
| | Incorrect | 424 | 40.8 |
| | Correct | 616 | 59.2 |
| What are ways to prevent the spread of mpox? | | | |
| | Incorrect | 357 | 34.3 |
| | Correct | 683 | 65.7 |
| Common symptoms of mpox disease are? | | - | |
| | Incorrect | 372 | 35.8 |
| | Correct | 668 | 64.2 |
| The case fatality ratio of mpox has historically ranged from? | | | |
| | Incorrect | 471 | 45.3 |
| | Correct | 569 | 54.7 |
| Preferred laboratory test for mpox diagnosis? | | | |
| | Incorrect | 543 | 52.2 |
| | Correct | 497 | 47.8 |
| Is there any vaccine available for mpox? | | | |
| | Incorrect | 728 | 70.0 |
| | Correct | 312 | 30.0 |

Knowledge section was assessed by giving a score of 1 to correct answer and 0 to wrong answer. A score of greater than equal to 7 was regarded as good knowledge (N = 358{34.4%}).

50,000–100,000 Pakistani Rupees (PKR) {225–450 USD} had lower odds of positive attitudes compared to participants who had an income of more than 100,000 PKR {450 USD}, (aOR = 0.593; 95% CI = 0.384–0.917, p = 0.019), while those aged 31–40 years had a more positive attitude (aOR = 2.454; 95% CI = 1.099–5.483, p = 0.029) (Table 6). With regards to good practices, males had lower odds of good practices than females (aOR = 0.741; 95% CI = 0.572–0.959, p = 0.023).

## Discussion

To the best of our knowledge, this is the first study carried out in Pakistan to assess the KAP towards MPXV among the general population. Based on our results, most participants had poor knowledge and attitudes but positive practices toward MPXV. Gender and education had a significant impact on knowledge and practices. While monthly income status had a significant impact on attitudes.

**Table 3. Attitude of general population regarding Mpox virus (N = 1040).**

| Variables | SD | D | N | A | SA | Average |
|---|---|---|---|---|---|---|
| Do you believe that mpox is a fatal disease? | 16 (1.5) | 105 (10.1) | 179 (17.2) | 505 (48.6) | 235 (22.6) | 3.81 |
| Do you believe that you are at risk of getting the disease? | 68 (6.5) | 190 (18.3) | 333 (32.0) | 351 (33.8) | 98 (9.4) | 3.21 |
| Do you believe that mpox can cause a pandemic like COVID-19? | 49 (4.7) | 246 (23.7) | 218 (21.0) | 398 (38.3) | 129 (12.4) | 3.30 |
| Do you believe that there should be a lockdown in areas reporting cases of mpox disease? | 44 (4.2) | 175 (16.8) | 211 (20.3) | 422 (40.6) | 188 (18.1) | 3.51 |
| Do you believe that travelling from endemic areas (such as Africa, Nigeria,etc.) to non-endemic areas should be banned? | 24 (2.3) | 136 (13.1) | 189 (18.2) | 464 (44.6) | 227 (21.8) | 3.71 |
| Do you believe that there is an increased risk of people getting the disease during Eid-ul-Adha? | 30 (2.9) | 136 (13.1) | 214 (20.6) | 478 (46.0) | 182 (17.5) | 3.62 |
| Do you believe that small pox vaccine is effective against mpox? | 33 (3.2) | 128 (12.3) | 377 (36.3) | 400 (38.5) | 102 (9.8) | 3.39 |
| **Overall attitude** | | | | | | |
| | Negative (7–25) | | | 606 (58.3%) | | |
| | Positive (26–35) | | | 434 (41.7%) | | |

Data presented as frequency (percentage), and average scores (mean) per total participants. SD = strongly disagree, D = Disagree, N = Neutral, A = Agree, SA = strongly agree. Attitude section was assessed by giving a score of 1 to strongly disagree and 5 to strongly agree. A score of greater than or equal to 26 was regarded as positive attitude.

Our study findings indicate that most study participants (65.6%) had poor knowledge about MPXV. This inference is consistent with other recent research that revealed that people generally in Saudi Arabia, Bangladesh, and the Kurdistan region of Iraq had little knowledge of MPXV [16, 36, 37]. As MPXV is endemic in tropical rainforest regions, Pakistan lies in a temperate climate zone, and there has been no previous exposure to the disease in the country, rendering the disease unknown among the general population [12, 38]. The majority were unaware of the origin of MPXV in the DRC. People visiting the endemic regions should be aware of the locally endemic diseases, which will help in the adoption of different preventive and treatment strategies, such as MPXV and COVID-19, which are endemic in the DRC and China, respectively [39, 40]. Thus, it is necessary to initiate awareness campaigns in Pakistan,

**Table 4. Practices of general population regarding Mpox virus (N = 1040).**

| Variables | SD | D | N | A | SA | Average |
|---|---|---|---|---|---|---|
| Will you take precautionary measures if mpox disease becomes an epidemic in your area? | 135 (13.0) | 80 (7.7) | 98 (9.4) | 383 (36.8) | 344 (33.1) | 3.69 |
| Quarantine is one way to prevent its spread at population levels just like COVID-19? | 94 (9.0) | 123 (11.8) | 239 (23.0) | 456 (43.8) | 128 (12.3) | 3.38 |
| Infected patients should be isolate from others? | 85 (8.2) | 56 (5.4) | 113 (10.9) | 499 (48.0) | 287 (27.6) | 3.81 |
| Avoiding crowds, using facemask and Washing hands frequently will prevent its spread? | 84 (8.1) | 74 (7.1) | 178 (17.1) | 508 (48.8) | 196 (18.8) | 3.63 |
| Do you believe that there is a need of more awareness in general public about mpox? | 90 (8.7) | 59 (5.7) | 99 (9.5) | 387 (37.2) | 405 (38.9) | 3.92 |
| Do you think government of Pakistan is taking sufficient steps to tackle with mpox? | 174 (16.7) | 332 (31.9) | 281 (27.0) | 195 (18.8) | 58 (5.6) | 2.64 |
| **Overall practice** | | | | | | |
| | Poor (6–21) | | | 440 (42.3%) | | |
| | Good (22–30) | | | 600 (57.7%) | | |

Data presented as frequency (percentage), and average scores (mean) per total participants. SD = strongly disagree, D = Disagree, N = Neutral, A = Agree, SA = strongly agree. Practice section was assessed by giving a score of 1 to strongly disagree and 5 to strongly agree. A score of greater than or equal to 22 was regarded as good practice.

**Table 5. Multivariate logistic regression for good knowledge of study participants.**

| Variables | Knowledge | | OR | 95% CI | P-Value | aOR | 95% CI | P-Value |
|---|---|---|---|---|---|---|---|---|
| | Poor knowledge | Good knowledge | | | | | | |
| **Gender** | | | | | | | | |
| Male | 403 (63.1) | 236 (36.9) | 1.339 | 1.026–1.748 | 0.032* | 1.324 | 1.001–1.752 | 0.049* |
| Female | 279 (69.6) | 122 (30.4) | Ref | - | - | Ref | - | - |
| **Age** | | | | | | | | |
| <20 | 154 (65.8) | 80 (34.2) | 1.732 | 0.784–3.824 | 0.174 | 1.769 | 0.765–4.090 | 0.182 |
| 21–30 | 423 (65.8) | 220 (34.2) | 1.734 | 0.809–3.716 | 0.157 | 1.667 | 0.751–3.699 | 0.209 |
| 31–40 | 75 (60.5) | 49 (39.5) | 2.178 | 0.952–4.981 | 0.065 | 1.853 | 0.793–4.330 | 0.154 |
| >40 | 30 (76.9) | 9 (23.1) | Ref | - | - | Ref | - | - |
| **Marital Status** | | | | | | | | |
| Married | 138 (64.8) | 75 (35.2) | 1.045 | 0.762–1.433 | 0.786 | - | - | - |
| Unmarried | 544 (65.8) | 283 (34.2) | Ref | - | - | - | - | - |
| **Residence** | | | | | | | | |
| Rural | 154 (61.6) | 97 (27.1) | 1.274 | 0.949–1.710 | 0.106 | 1.214 | 0.892–1.654 | 0.218 |
| Urban | 528 (66.9) | 261 (33.1) | Ref | - | - | Ref | - | - |
| **Education** | | | | | | | | |
| Primary up to Matriculation | 34 (89.5) | 4 (10.5) | 0.213 | 0.073–0.626 | 0.005* | 0.220 | 0.074–0.653 | 0.006* |
| Intermediate | 127 (64.5) | 70 (35.5) | Ref | - | - | Ref | - | - |
| Graduation | 396 (66.6) | 199 (33.4) | 0.912 | 0.650–1.278 | 0.592 | 0.923 | 0.637–1.337 | 0.671 |
| Post-graduation | 125 (59.5) | 85 (40.5) | 1.234 | 0.826–1.843 | 0.305 | 1.287 | 0.801–2.067 | 0.296 |
| **Monthly income in PKR (USD)** | | | | | | | | |
| <50000 (225) | 436 (67.7) | 208 (32.3) | 0.897 | 0.613–1.312 | 0.575 | - | - | - |
| 50000–100,000 (225–450) | 152 (60.3) | 100 (39.7) | 1.237 | 0.808–1.894 | 0.328 | - | - | - |
| >100,000 (450) | 94 (65.3) | 50 (34.7) | Ref | - | - | - | - | - |

OR = Odds Ratio, CI = Confidence interval.

A p value of less than 0.05 considered statistically significant.

especially for professional travelers, the business community, and tourists to the endemic region, to prevent the spread of disease to non-endemic areas.

However, participants in this study had good knowledge about the causative factor of mpox fever (87.0%), prevention of the disease (65.0%), and common symptoms (64%) (Table 7). Except for a few variations, smallpox, and mpox viruses belong to the same family, and many similarities are found in terms of disease presentation, spread, prevention, control, and management plans [41–43]. This might be one reason for being known common symptomatology of MPXV infection and preventive measures. But only 30% knew that the smallpox vaccine is also effective against MPXV. Vaccine against smallpox is considered beneficial for the prevention and control of MPXV [44, 45]. Some analyses also depicted that the MPXV outbreak was due to a break in vaccination against smallpox [46, 47].

Our study shows that most participants (58.3%) had a negative attitude toward MPXV, which is consistent with a study conducted on the general public's attitude toward mpox in the United States [48]. Good knowledge is not beneficial without applying that knowledge in practical life. Individual knowledge can be practically utilized to adopt a positive attitude by engaging people in awareness activities. Studies in Iran and Indonesia about avian influenza postulated similar aspects of the correlation of knowledge with attitudes [49, 50]. Most respondents believed that MPXV is a fatal disease that can cause a pandemic like COVID-19. The COVID-19 pandemic hit many people hrad by evolving a lot of complications for them, and

**Table 6. Multivariate logistic regression analysis for positive attitude of study participants.**

| Variables | Attitude | | OR | 95% CI | P-Value | aOR | 95% CI | P-Value |
|---|---|---|---|---|---|---|---|---|
| | Negative | Positive | | | | | | |
| **Gender** | | | | | | | | |
| Male | 375 (58.7) | 264 (41.3) | 0.957 | 0.743–1.232 | 0.731 | - | - | - |
| Female | 231 (57.6) | 170 (42.4) | Ref | - | - | - | - | - |
| **Age** | | | | | | | | |
| <20 | 142 (60.7) | 92 (39.3) | 1.649 | 0.783–3.474 | 0.188 | 1.390 | 0.634–3.049 | 0.411 |
| 21–30 | 369 (57.4) | 274 (42.6) | 1.890 | 0.925–3.863 | 0.081 | 1.900 | 0.895–4.035 | 0.095 |
| 31–40 | 67 (54.0) | 57 (46.0) | 2.166 | 0.991–4.732 | 0.053 | 2.454 | 1.099–5.483 | 0.029* |
| >40 | 28 (71.8) | 11 (28.2) | Ref | - | - | Ref | - | - |
| **Marital Status** | | | | | | | | |
| Married | 124 (58.2) | 89 (41.8) | 1.003 | 0.739–1.361 | 0.986 | - | - | - |
| Unmarried | 482 (58.3) | 345 (41.7) | Ref | - | - | - | - | - |
| **Residence** | | | | | | | | |
| Rural | 134 (53.4) | 117 (46.6) | 1.300 | 0.977–1.730 | 0.072 | 1.266 | 0.945–1.697 | 0.114 |
| Urban | 472 (59.8) | 317 (40.2) | Ref | - | - | Ref | - | - |
| **Education** | | | | | | | | |
| Primary up to Matriculation | 20 (52.6) | 18 (47.4) | 1.048 | 0.523–2.102 | 0.894 | 0.987 | 0.478–2.037 | 0.972 |
| Intermediate | 106 (53.8) | 91 (46.2) | Ref | - | - | Ref | - | - |
| Graduation | 355 (59.7) | 240 (40.3) | 0.787 | 0.569–1.089 | 0.149 | 0.710 | 0.495–1.018 | 0.062 |
| Post-graduation | 125 (59.5) | 85 (40.5) | 0.792 | 0.535–1.173 | 0.245 | 0.787 | 0.489–1.265 | 0.322 |
| **Monthly income in PKR** (USD) | | | | | | | | |
| <50000 (225) | 348 (54.0) | 296 (46.0) | 1.191 | 0.826–1.717 | 0.349 | 1.145 | 0.777–1.686 | 0.493 |
| 50000–100,000 (225–450) | 174 (69.0) | 78 (31.0) | 0.628 | 0.410–0.960 | 0.032* | 0.593 | 0.384–0.917 | 0.019* |
| >100,000 (450) | 84 (58.3) | 60 (41.7) | Ref | - | - | Ref | - | - |

OR = Odds Ratio, aOR = Adjusted Odds Ratio, CI = Confidence interval.

A p value of less than 0.05 considered statistically significant.

based on this, it might be possible they are considering MPXV to be fatal and a pandemic in the future [51, 52]. Some studies depict the role of the fatality of disease over one's psychology as being the case with the Ebola virus disease [53, 54]. However, little information is available regarding the connection between MPXV and psychological issues [55].

In our study, most respondents had good practices compared to knowledge and attitudes. The findings are consistent with the studies conducted on general physicians showing good practices [32, 56]. In our study, most participants believed that infected patients should be isolated from others and that preventive measures such as face masks, hand washing, and quarantine would prevent the spread of MPXV. This might be due to the recent pandemic of COVID-19, during which taking precautionary measures was the sole mode of protection all over the world before the vaccine's introduction [57]. The other reason for good practices could be recent alerts from WHO and CDC.

In agreement with the previous study [36], our study reported significantly good knowledge of MPXV among male participants. This may be connected to how MPXV is transmitted because the majority of cases in the most recent outbreak have mostly been reported among men who have had sex with men (MSM) [58, 59]. According to a recent pooled meta-analysis, sexual contact is involved in more than 91% of cases [6]. A study reported that all (100%) of the 54 MPX cases that presented at one health center in the UK were MSM [60]. Study findings affirm that respondents with higher education levels had better knowledge than those

**Table 7. Multivariate logistic regression analysis for good practices of study participants.**

| Variables: Knowledge about Mpox virus | Value: N(%) |
|---|---|
| **Mpox disease was first isolated in?** | |
| Nigeria | 269(25.9) |
| **Congo** | **496(47.7)** |
| China | 224(21.5) |
| Japan | 51(4.9) |
| **The causative factor of mpox disease is?** | |
| Bacteria | 83(8.0) |
| **Virus** | **905(87.0)** |
| Fungus | 33(3.2) |
| Parasite | 19(1.8) |
| **Transmission of mpox disease occur by?** | |
| Human-to-human | 200(19.2) |
| Animal-to-human | 226(21.7) |
| Contact with contaminated material | 65(6.3) |
| **All** | **549(52.8)** |
| **Human-to-human transmission occurs by?** | |
| Respiratory secretions of an infected person or animal | 220(21.2) |
| Skin lesions of an infected person | 227(21.8) |
| Contact with contaminated objects | 78(7.5) |
| Feces or urine of infected person or animal | 29(2.8) |
| **All** | **486(46.7)** |
| **The incubation period (interval from infection to onset of symptoms)?** | |
| 1–5 days | 237(22.8) |
| **6-13days** | **616(59.2)** |
| 14-30days | 172(16.5) |
| 1–2 months | 15(1.4) |
| **What are ways to prevent the spread of mpox?** | |
| Avoid contact with animals that could harbor the virus | 109(10.5) |
| Practice good hand hygiene after contact with infected animals or humans. | 161(15.5) |
| Isolate infected patients from others | 87(8.4) |
| **All** | **683(65.7)** |
| **Common symptoms of mpox disease?** | |
| Loose motions, abdominal pain, and vomiting | 159(15.3) |
| **Fever, headache, swelling of the lymph nodes, and muscle aches** | **668(64.2)** |
| Fever, cough and shortness of breath | 120(11.5) |
| Fever, sore throat and cough | 93(8.9) |
| **The case fatality ratio of mpox has historically ranged from?** | |
| **0–11%** | **569(54.7)** |
| 15–30% | 310(29.8) |
| 40–50% | 108(10.4) |
| More than 70% | 53(5.1) |
| **Preferred laboratory test for mpox diagnosis?** | |
| Serology | 165(15.9) |
| Complete blood test (CBC) | 206(19.8) |
| Diagnosis is clinical, Need no laboratory confirmation | 172(16.5) |
| **PCR (Polymerase chain reaction)** | **497(47.8)** |
| **Is there any vaccine available for mpox?** | |

*(Continued)*

**Table 7.** (Continued)

| Smallpox vaccine is also effective against mpox | 312(13.1) |
|---|---|
| Vaccine against mpox is available. | 273(26.3) |
| No vaccine is available | 455(43.8) |

OR = Odds Ratio, aOR = Adjusted Odds Ratio, CI = Confidence interval.

A p value of less than 0.05 considered statistically significant.

participants with qualifications of primary to matriculation (Class-10). Other studies with similar findings also revealed that a greater level of education was related to having good knowledge [61, 62]. Education has a key role in promoting healthy behaviors and gaining control over epidemics, by improving knowledge levels about any health-related case. Probably, this is because of more exposure to media, internet, and newspapers that provide easy access to knowledge and a higher maturity level. However, the association of age, residence, and monthly income with knowledge did not withstand the regression analysis in the current study.

Nevertheless, our results showed that the population with a low income had a more positive attitude. This is not consistent with many studies that revealed having a higher socioeconomic position corresponds with having a more positive attitude [63, 64]. People with higher socioeconomic status have additional opportunities for higher education, residing in urban areas, and exposure to mass media. The study findings demonstrate that females were shown to have more good practices compared to males. Similar findings were observed in other studies showing that females take more preventive measures than males [65]. Women claimed to be more responsible for their health, potentially related to risk perception and the gender bias that women are socialized to be more concerned about health issues than men.

Mpox is an emerging virulent disease poses a serious threat to worldwide public health [42]. Given the substantial risks posed by MPXV and the lack of mpox vaccine, prevention strategies play a crucial role in minimizing infection rates and halting the disease's spread. It has been acknowledged that increasing awareness through campaigns and seminars, as well as embracing preventive strategies can halt infectious diseases like COVID-19, influenza, and dengue from becoming epidemics or pandemics [66–68]. Studies conducted to assess knowledge and attitudes about disease prevention and improve public health interventions are futile without public interest and collaboration [32, 69]. This highlights how crucial it is for the general public to follow preventative and control measures based on their knowledge, attitudes, and practice (KAP).

## Study strengths and limitations

While interpreting the study's findings, some limitations should be considered. First, there was a possibility of selection bias because the convenience sampling method was used. Second, the questionnaire was distributed through an online system using different social media platforms. Consequently, there is a possibility of bias as underprivileged populations would not have been able to participate, and results cannot be extended to the entire community. Third, this study is based on self-reported data, which might lead to internet surfing to acquire information for a difficult and technical question in the survey, resulting in a biased response to some questions. According to what they believe to be expected, participants may have responded positively to attitude and practice questions. Fourth, a recall bias during the survey cannot be ignored. Finally, results stratified by marital status and residence should be

interpreted with caution, since the majority of the participants were unmarried and urban. However, the current study was strengthened by the large sample size. To the researchers' knowledge, this is the first study analyzing KAP toward MPXV, in the Pakistan's public. Therefore, the results of this study may help health authorities to implement effective policies to combat the emergence of MPXV in Pakistan.

## Conclusion

In summary, the present study demonstrated inadequate knowledge and attitudes regarding MPXV among the Pakistani population. Our results showed that good knowledge could result in good attitudes and practices, which are crucial to minimizing the expanding impacts of the disease. The current outbreak of MPXV in non-endemic countries warrants strict epidemiological surveillance to restrict the further dissemination of the outbreak in other non-endemic countries. International organizations, national health authorities, national organizations, health care workers, and the media sector must be actively involved in the implementation of appropriate interventions for the awareness and prevention of MPXV.

## Supporting information

**S1 File.**
(SAV)

## Author Contributions

**Conceptualization:** Hashaam Jamil.

**Data curation:** Kashmala Idrees, Samra Akram.

**Formal analysis:** Hashaam Jamil, Muhammad Sohaib Asghar.

**Software:** Muhammad Sohaib Asghar.

**Supervision:** Ranjit Sah.

**Validation:** Muhammad Junaid Tahir.

**Visualization:** Ali Ahmed.

**Writing – original draft:** Muhammad Idrees, Waleed Tariq, Qudsia Sayyeda.

**Writing – review & editing:** Muhammad Junaid Tahir, Kaleem Ullah, Ali Ahmed, Aroop Mohanty, Bijaya Kumar Padhi, Ranjit Sah.

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
