## [Decision Letter · Decision Letter 0]

24 Feb 2023

PONE-D-22-34623Socio-demographic determinants of monkeypox virus preventive behavior: a cross-sectional study in PakistanPLOS ONE

Dear Dr. Sah,

Thank you for submitting your manuscript to PLOS ONE. After careful consideration, we feel that it has merit but does not fully meet PLOS ONE’s publication criteria as it currently stands. Therefore, we invite you to submit a revised version of the manuscript that addresses the points raised during the review process.

ACADEMIC EDITOR:

The manuscript will be re-evaluated after major revisions as suggested by the reviewers.

We look forward to receiving your revised manuscript.

Kind regards,

Om Prakash Choudhary, Ph.D.

Academic Editor

PLOS ONE

Journal Requirements:

3. We note you have included a table to which you do not refer in the text of your manuscript. Please ensure that you refer to Table 7 in your text; if accepted, production will need this reference to link the reader to the Table.

Reviewers' comments:

Reviewer's Responses to Questions

**Comments to the Author**

1. Is the manuscript technically sound, and do the data support the conclusions?

Reviewer #1: Yes

Reviewer #2: Yes

2. Has the statistical analysis been performed appropriately and rigorously? 

Reviewer #1: Yes

Reviewer #2: Yes

3. Have the authors made all data underlying the findings in their manuscript fully available?

Reviewer #1: Yes

Reviewer #2: Yes

4. Is the manuscript presented in an intelligible fashion and written in standard English?

Reviewer #1: Yes

Reviewer #2: No

5. Review Comments to the Author

Reviewer #1: The manuscript is written well, however, some of the points need to be addressed before its publication. I have mentioned all the points below:

1. Some of the references are not as per the format of the journal. Kindly update the same in the revisions.

2. Read the complete manuscript because some spelling mistakes.

3. There are a few grammatical mistakes in the manuscript, hence I request authors check the manuscript carefully before submitting the revised version of the manuscript

Rest is ok.

Reviewer #2: I have gone through the manuscript, it contains adequate information. However, the English is not satisfactory, serious language editing is required. Sections are well defined but major comments are needed to address to improve this manuscripts.

1. Major language issues found that needs expert attention. Some sentences could be more precise. It seems to be some words that are overused along with article usage issues. Some sentences should be rewritten. The authors should check the punctuation marks because some are missing from the right places.

2. The present title of the article looks proper and concise.

3. The current abstract is concise and focused. Nevertheless, there are also some article (a, an, the) and punctuation usage problems.

4. Please use italics in the introduction in the genus, species names, and genes.

5. Keyword looks good and adequate.

6. Please follow the journal rules in every major section and subsection; every subsection within major sections will be Level 2 headings. It will be in bold and 16pt font. It is suggested to check the subsections of the Method and result sections. Do not use italic unnecessarily. Only use it in genus and species names, genes, etc. All level 1 heading will be bold and in 18pt font.

7. In the discussion part, Line 266, there is an error in writing. Understand that it could be a typographic problem. So, it ‘that’ should be removed.

8. The tables are clearly labeled, but they need to be ideally positioned. It should be close to the relevant text after the paragraph in which they are first cited.

9. Found some issues in reference citations; please cite references in brackets. For example, “[1], [2-5], or [3, 7, 9].

10. As per journal requirements, Authors should list the supporting information captions at the end of the manuscript in a section titled “Supporting information.” Use Level 1 and bold type for the titles.

11. In line no 303, a letter needs to be included.

6. PLOS authors have the option to publish the peer review history of their article (what does this mean?). If published, this will include your full peer review and any attached files.

Reviewer #1: No

Reviewer #2: No

---

## [Author Response · Author response to Decision Letter 0]

10 Mar 2023

Dear Editor, 

PLOS ONE

We would like to thank you for considering our manuscript ‘Socio-demographic determinants of monkeypox virus preventive behavior: a cross-sectional study in Pakistan’ and for the valuable feedback provided. 

We are very grateful for the reviews provided by the editors and each of the external reviewers of this manuscript. The comments are encouraging, and the reviewers appear to share our judgment that this study and its results are important. Please find our detailed response to the comments below in blue. All page numbers refer to the manuscript file with tracked changes. We have also attached the clean files without track changes.

We hope the paper will now be acceptable for publication in the PLOS ONE.

Yours Sincerely,

Corresponding Author

Responses to Reviewer 1

Reviewer’s comment: Some of the references are not as per the format of the journal. Kindly update the same in the revisions.

Authors’ response: Dear reviewer, thank you for valuable correction. The changes have been done throughout the manuscript as per your suggestions.

Reviewer’s comment: Read the complete manuscript because some spelling mistakes.

Authors’ response: We have revised the manuscript, and corrected spelling mistakes.

Reviewer’s comment: There are a few grammatical mistakes in the manuscript, hence I request authors check the manuscript carefully before submitting the revised version of the manuscript

Authors’ response: We have checked the grammatical mistakes and some corrections have been done.

Responses to Reviewer 2

Reviewer’s comment: Major language issues found that needs expert attention. Some sentences could be more precise. It seems to be some words that are overused along with article usage issues. Some sentences should be rewritten. The authors should check the punctuation marks because some are missing from the right places.

Authors’ response: Dear reviewer, thanks for the comment. We have tried to improve English language.

Reviewer’s comment: The present title of the article looks proper and concise.

Authors’ response: Dear reviewer, Thank you for your appreciation.

Reviewer’s comment: The current abstract is concise and focused. Nevertheless, there are also some article (a, an, the) and punctuation usage problems.

Authors’ response: We have revised the abstract and made some corrections accordingly.

Reviewer’s comment: Please use italics in the introduction in the genus, species names, and genes.

Authors’ response: Thanks for correction. [Addressed in Page 03; Line No.63-64]

Reviewer’s comment: Keyword looks good and adequate.

Authors’ response: Dear reviewer, Thank you for your appreciation.

Reviewer’s comment: Please follow the journal rules in every major section and subsection; every subsection within major sections will be Level 2 headings. It will be in bold and 16pt font. It is suggested to check the subsections of the Method and result sections. Do not use italic unnecessarily. Only use it in genus and species names, genes, etc. All level 1 heading will be bold and in 18pt font.

Authors’ response: Correction has been done. Now, we have used bold and 18pt font level 1 headings. Bold and 16pt font level 2 headings.

Reviewer’s comment: In the discussion part, Line 266, there is an error in writing. Understand that it could be a typographic problem. So, it ‘that’ should be removed.

Authors’ response: Thanks for correction. The word ‘that’ has been removed now. [Addressed in Page 11; Line No.277]

Reviewer’s comment: The tables are clearly labeled, but they need to be ideally positioned. It should be close to the relevant text after the paragraph in which they are first cited.

Authors’ response: Thanks for correction. The tables are positioned as per your recommendation.

Reviewer’s comment: Found some issues in reference citations; please cite references in brackets. For example, “[1], [2-5], or [3, 7, 9].

Authors’ response: The changes have been done throughout the manuscript as per journal requirements.

Reviewer’s comment: As per journal requirements, Authors should list the supporting information captions at the end of the manuscript in a section titled “Supporting information.” Use Level 1 and bold type for the titles.

Authors’ response: The change has been done as per your suggestion.

Reviewer’s comment: In line no 303, a letter needs to be included

Authors’ response: The change has been done as per your suggestion. [Addressed in Page 19; Line No.373]

---

## [Decision Letter · Decision Letter 1]

29 Mar 2023

Socio-demographic determinants of monkeypox virus preventive behavior: a cross-sectional study in Pakistan

PONE-D-22-34623R1

Dear Dr. Sah,

We’re pleased to inform you that your manuscript has been judged scientifically suitable for publication and will be formally accepted for publication once it meets all outstanding technical requirements.

Kind regards,

Jan Rychtář

Academic Editor

PLOS ONE

Additional Editor Comments (optional):

Based on the recommendation of the two reviewers, the paper is now acceptable for publication in PLOS ONE.

Reviewers' comments:

Reviewer's Responses to Questions

**Comments to the Author**

1. If the authors have adequately addressed your comments raised in a previous round of review and you feel that this manuscript is now acceptable for publication, you may indicate that here to bypass the “Comments to the Author” section, enter your conflict of interest statement in the “Confidential to Editor” section, and submit your "Accept" recommendation.

Reviewer #1: All comments have been addressed

Reviewer #2: All comments have been addressed

2. Is the manuscript technically sound, and do the data support the conclusions?

Reviewer #1: Yes

Reviewer #2: Yes

3. Has the statistical analysis been performed appropriately and rigorously? 

Reviewer #1: Yes

Reviewer #2: Yes

4. Have the authors made all data underlying the findings in their manuscript fully available?

Reviewer #1: Yes

Reviewer #2: Yes

5. Is the manuscript presented in an intelligible fashion and written in standard English?

Reviewer #1: Yes

Reviewer #2: Yes

6. Review Comments to the Author

Reviewer #1: accept the manuscript

Reviewer #2: This is a very informative research article that is well-written. The authors utilizes tables and figures effectively, which is one of its strengths. The topic of this article is appropriate and explains the scientific findings properly. The English and manuscript preparation are appropriate for publication. This article's writing is pretty good and the presentation is positive overall. The sections are thorough and clearly stated and previous issues have been improved.

7. PLOS authors have the option to publish the peer review history of their article (what does this mean?). If published, this will include your full peer review and any attached files.

Reviewer #1: No

Reviewer #2: No

---

## [Editor Report · Acceptance letter]

13 Apr 2023

PONE-D-22-34623R1 

Socio-demographic determinants of Monkeypox virus preventive behavior: a cross-sectional study in Pakistan 

Dear Dr. Sah:

I'm pleased to inform you that your manuscript has been deemed suitable for publication in PLOS ONE. Congratulations! Your manuscript is now with our production department. 

Kind regards, 

on behalf of

Dr. Jan Rychtář 

Academic Editor

PLOS ONE